

# Serum heparan sulfate and chondroitin sulfate concentrations in patients with newly diagnosed exfoliative glaucoma

Vesna D. Maric[1,2], Marija M. Bozic[1,2], Andja M. Cirkovic[3], Sanja Dj Stankovic[4], Ivan S. Marjanovic[1,2] and Anita D. Grgurevic[5]

[1] Clinic for Eye Diseases, Clinical Center of Serbia, Belgrade, Serbia
[2] Faculty of Medicine, University of Belgrade, Belgrade, Serbia
[3] Department for Medical Statistics and Informatics, Faculty of Medicine, University of Belgrade, Belgrade, Serbia
[4] Center for Medical Biochemistry, Clinical Center of Serbia, Belgrade, Serbia
[5] Institute of Epidemiology, Faculty of Medicine, University of Belgrade, Belgrade, Serbia

Corresponding author
Vesna D. Maric,
vesbabic@sezampro.rs

## ABSTRACT

**Background:** Exfoliative glaucoma (XFG) is typically classified as a high-pressure type of secondary open-angle glaucoma that develops as a consequence of exfoliation syndrome (XFS). Exfoliation syndrome is an age-related, generalized disorder of the extracellular matrix characterized by production and progressive accumulation of a fibrillar exfoliation material (XFM) in intra- and extraocular tissues. Exfoliation material represents complex glycoprotein/proteoglycan structure composed of a protein core surrounded by glycosaminoglycans such as heparan sulfate (HS) and chondroitin sulfate (CS). The purpose of the present study was to investigate HS and CS concentrations in serum samples of patients with newly diagnosed XFG and compare the obtained values with those pertaining to newly diagnosed primary open-angle glaucoma (POAG), normal controls (NC) and subjects with XFS.

**Methods:** This case–control study involved 165 subjects, including patients with newly diagnosed XFG, patients with newly diagnosed POAG, subjects with XFS and age- and sex-matched NC. The study was conducted at the Glaucoma Department of Clinic for Eye Diseases, Clinical Centre of Serbia, as the referral center for glaucoma in Serbia.

**Results:** The mean age in the XFG, POAG, XFS and NC groups was $73.3 \pm 9.0$, $66.3 \pm 7.8$, $75.5 \pm 7.0$ and $73.5 \pm 9.5$ years, respectively, XFG vs. POAG, $p < 0.001$. Mean serum HS concentrations in the XFG, POAG, NC and XFS groups were $3,189.0 \pm 1,473.8$ ng/mL, $2,091.5 \pm 940.9$ ng/mL, $2,543.1 \pm 1,397.3$ ng/mL and $2,658.2 \pm 1,426.8$ ng/mL respectively, XFG vs. POAG, $p = 0.001$ and XFG vs. NC, $p = 0.032$. Mean serum CS concentrations in the XFG, POAG, NC and XFS group were $43.9 \pm 20.7$ ng/mL, $38.5 \pm 22.0$ ng/mL, $35.8 \pm 16.4$ ng/mL and $43.3 \pm 21.8$ ng/mL, respectively, XFG vs. NC, $p = 0.041$.

**Conclusions:** Our findings revealed greater HS and CS concentrations in XFG patients and XFS subjects compared to those without XFM. Implications of HS and CS in the pathophysiology of XFS and glaucoma should be studied further. Serum is easily accessible and should thus be explored as rich sources of potential biomarkers. Further research should aim to identify XFG biomarkers that could be utilized in routine blood analysis tests, aiding in timely disease diagnosis.

# INTRODUCTION

Glaucoma is a term describing a group of ocular disorders with multi-factorial etiology and it is a form of optic neuropathy related to elevated intraocular pressure as the main risk factor (*Weinreb, Aung & Medeiros, 2014*). It is the second leading cause of blindness worldwide (*Quigley & Broman, 2006*), with primary open-angle glaucoma (POAG) and exfoliative glaucoma (XFG) as its most prevalent types in developed countries (*Prum et al., 2016*; *Hollo, Katsanos & Konstas, 2015*).

Exfoliation syndrome (XFS) is an age-related, generalized disorder of the extracellular matrix (ECM) characterized by production and progressive accumulation of a fibrillar exfoliation material (XFM) deposits in tissues throughout the anterior segment, as well as in connective tissues comprising various visceral organs (*Ritch, Schlötzer-Schrehardt & Konstas, 2003*; *Ritch & Schlötzer-Schrehardt, 2001*). The most widely recognized disease manifestation is XFG, while glaucoma develops in only about 30% of XFS eyes in the patients' lifetime (*Ritch, 2015*).

Exfoliative glaucoma is typically classified as a high-pressure type of secondary open-angle glaucoma (*Puska, 2015*). Compared with POAG, more adverse clinical progression and poorer prognosis is usually associated with XFG (*Ritch & Schlötzer-Schrehardt, 2001*).

According to the findings yielded by immunohistochemical and biochemical studies, XFM represents a highly glycosylated, cross-linked and enzymatically resistant glycoprotein/ proteoglycan complex (*Zenkel & Schlötzer-Schrehardt, 2014*; *Schlötzer-Schrehardt & Naumann, 2015*). Proteoglycans consist of a core protein to which glycosaminoglycan (GAG) side chains are attached. Glycosaminoglycans are an important component of the XFM. They are linear polysaccharides comprised of glucosamine-/ galactosamine-containing repeating disaccharides (*Yue, 2014*). There are four GAG groups: hyaluronic acid, keratan sulfate, chondroitin/dermatan sulfate and heparan sulfate (HS), including heparin (*Huflejt et al., 2014*).

In both XFS and XFG, the progressively accumulating XFM successively detaches, destroys and replaces the normal ECM, e.g., the basement membranes of the cells, and ultimately results in degeneration of the cells involved (*Schlötzer-Schrehardt & Naumann, 2006*; *Schlötzer-Schrehardt, 2012*). Thus, overproduction and abnormal metabolism of GAGs has been suggested as one of the key changes in XFS (*Baba, 1983*; *Schlötzer-Schrehardt, Dorfler & Naumann, 1992*). As noted above, HS and chondroitin sulfate (CS) are two major GAG types (*Lu et al., 2010*; *Li et al., 2017*). Therefore, ability to quantify changes in blood GAG structures will advance the understanding and diagnosis of human diseases (*Lu et al., 2010*).

The purpose of the present study was to investigate HS and CS concentrations in serum samples of patients with newly diagnosed untreated XFG and compare the obtained values with those pertaining to newly diagnosed untreated POAG, normal controls (NC) and subjects with XFS. The second goal was to investigate the relationship

between the HS and CS serum concentrations and clinical parameters in XFG, POAG and XFS groups.

## MATERIALS AND METHODS

### Study population

This case–control study involved 165 consecutive patients that were seen between June 2016 and December 2017 at the Glaucoma Department of Clinic for Eye Diseases, Clinical Centre of Serbia, as the referral center for glaucoma in Serbia. The sample comprised of patients with newly diagnosed XFG and age- and sex-matched NC, along with patients with newly diagnosed POAG and subjects with XFS. Subjects were classified as having POAG if they presented with the typical glaucomatous optic disc (neural rim thinning or notching, saucerization, thin nasal rim or total cupping) and/or glaucoma visual field changes, in the presence of an IOP $\geq$ 22 mm Hg without medication, and a gonioscopy finding of a wide and open anterior chamber angle. Exfoliative glaucoma was diagnosed based on typical glaucomatous optic disc and/or glaucoma visual field changes in the presence of an IOP $\geq$ 22 mm Hg without medication, with presence of exfoliation on the pupil edge and/or the anterior lens capsule after mydriasis by biomicroscopic evaluation in either or both eyes. Finally, XFS diagnosis was established by visualization of XFM on the pupillary margin and/or on the anterior lens surface after pupillary dilation, along with an IOP < 22 mm Hg, in the absence of glaucomatous optic nerve damage and visual field changes. The patient was classified as having XFS if XFM was present in either or both eyes. Normal control subjects had no evidence of XFS or glaucoma based on clinical examination.

Exclusion criteria were: (1) use of anti-glaucoma medications; (2) use of topical/systemic steroids; (3) previous intraocular surgery; (4) history of ocular trauma, uveitis, corneal scars, lens-induced glaucoma, proliferative diabetic retinopathy and any other ocular pathology that could have led to secondary glaucoma; (5) diseases that could influence the HS and CS levels, such as any type of cancer or rheumatoid arthritis and late-stage osteoarthritis (*Li et al., 2017*; *Pothacharoen et al., 2006*). Anti-glaucoma medications were not used by 244 subjects, six of whom had intraocular surgery and 15 used topical and systemic steroids in the 6 months prior to joining the study. In addition, 35 of the remaining 223 subjects were excluded due to history of ocular trauma, uveitis, corneal scars, lens-induced glaucoma and proliferative diabetic retinopathy. Further 23 subjects were excluded from the study due to comorbidity that could influence the HS and CS levels: cancer (10), rheumatoid arthritis and late-stage osteoarthritis diseases (13). Thus, the final sample comprised of 165 subjects.

All subjects that met the study inclusion criteria received a detailed explanation of the study purpose and the nature of their involvement, and those that agreed to take part in the investigation signed an informed consent form, in accordance with the principles embodied in the Declaration of Helsinki. The study as a part of the doctoral dissertation was reviewed and approved by the Ethics Committee of the Faculty of Medicine, University of Belgrade, record number 29/III-3.

## Collecting data

Participants' demographic and comorbidity characteristics were obtained via interviews and by reviewing medical documentation. While age and gender were the only demographic data of interest, comorbidities included presence of systemic diseases, such as diabetes mellitus (DM), systemic hypertension, history of myocardial stroke, history of coronary artery bypass or vascular surgery, history of abdominal aortic aneurysm, arrhythmia and history of acute cerebrovascular disease.

## Eye examinations

Ocular examination in all patients was performed by one ophthalmologist (VM) and included visual acuity (VA), slit-lamp biomicroscopy, gonioscopy (using Goldmann two-mirror indirect gonioscope), IOP measurement (using Goldmann applanation tonometry) and dilated fundus examination (using Volk Superfield +90 D lens). The mean IOP based on three readings in each eye was adopted as the pressure for that eye. A visual field test was performed using the Threshold C 24-2 Swedish Interactive Testing Algorithm standard program with Humphrey Visual Field Analyzer II (Carl Zeiss, Germany). Visual acuity was measured by Snellen chart at six m distance and was converted to decimal notation, whereby the best-corrected visual acuity was recorded. If the patients were unable to read any letters displayed on the chart, their ability to count fingers was used as the VA. The next lower level of vision would be indicated by the ability to perceive light (denoted as "LP" or "light perception"). Complete blindness was diagnosed if no light perception ("NLP") was determined by clinical examination.

If the anterior chamber angle was open, participants had their pupils dilated by administering dilation drops containing 5% phenylephrine and 1% tropicamide. Prior to pupil dilation, a detailed high-magnification slit-lamp assessment of the pupil margin was carried out. After pupil dilation, the anterior lens surface in each eye was scanned, looking specifically for signs of XFM. If the angle was potentially occludable, the lens and the fundus evaluations were performed without dilation, and the participants were referred for a laser peripheral iridotomy. In these cases, dilated lens and fundus examinations were performed upon iridotomy completion.

Indices of glaucoma severity were expressed numerically as cup to disc ratio (C/D), allowing the vertical C/D (vC/D) to be reported, along with the staging of visual field defects using Hodapp Classification, and separately visual field mean deviation (MD) and pattern standard deviation (PSD).

## BLOOD SAMPLING

After eye examinations, blood samples were collected from each subject. Samples were collected into a serum separator tube. Blood was coagulated at room temperature for two hours and was subsequently centrifuged at approximately $1,000\times$ g for 15 min. Serum was separated into aliquots and was stored at $-70\ ^\circ$C until required for analyses.

## Assay

Serum human HS concentration was measured using commercial ELISA Kit (Cusabio, Houston, TX, USA). This assay employs the competitive inhibition enzyme immunoassay

technique and has a 20–8,000 ng/mL measurement range, with eight ng/mL detection limit. When using this ELISA Kit, intra-assay and inter-assay precision <6% and <11% is achieved, respectively. In this work, antibody specific to HS was pre-coated onto a microplate and standards and samples were pipetted into the wells with a horseradish perxidase (HRP) conjugated HS. A competitive inhibition reaction was launched between HS (standards or samples) and HRT-conjugated HS with the pre-coated antibody specific for HS. Following a wash to remove any unbound reagent, a substrate solution was added to the wells, whereby the developed color is inversely proportional to the amount of HS in the sample. Thus, when color development is terminated, color intensity is measured and is converted to the corresponding HS value.

Serum human CS concentration was measured using commercial ELISA Kit (Abbexa, Cambridge, UK). The measurement range was 3.13–200 ng/mL with 1.88 ng/mL detection limit. In this case, <8% and <10% intra-assay and inter-assay precision can be achieved, respectively. This kit is based on sandwich enzyme-linked immunosorbent assay technology. When using this technique, an antibody specific to CS is pre-coated onto a 96-well plate. The standards, test samples and biotin detection antibody are added to the wells and rinsed with wash buffer. Biotin conjugated CS antibody serves as a detection antibody, whereas 3,3′,5,5′-Tetramethylbenzidine (TMB) substrate is used to visualize HRP enzymatic reaction. TMB is catalyzed by HRP to produce a blue-colored product that changes into yellow after adding stop solution. The intensity of the color yellow is proportional to the CS amount bound on the plate. The OD absorbance is measured spectrophotometrically at 450 nm in a microplate reader, allowing the CS concentration to be calculated.

Blood (serum) glucose was determined for all subjects, all of whom were given the hemoglobin A1c (HbA1c) test, to determine the average blood sugar level over the preceding 3 months. Serum glucose was measured using the commercial assay on Roche Cobas 6000 automated analyzer (Roche Diagnostics, Mannheim, Germany). The measurement range was 3.9–6.1 mmol/L. Hemoglobin A1c analysis was performed using capillary electrophoresis technique in free solution on CapillaryFlex Piercing II instrument (Sebia, Lisses, France).

## Statistical analysis

Standard descriptive statistics were used (arithmetic mean with standard deviation for normally distributed numerical data, or median with range otherwise, while providing absolute and relative numbers for categorical data). Whether the data were distributed normally were established via mathematical (coefficient of variation, skewness and kurtosis, Kolmogorov–Smirnov and Shapiro–Wilk test) and graphical (histogram, normal Q-Q diagram, detrended Q-Q diagram and box-plot) methods. As at least two of the mathematical as well as graphical tests met the normal distribution criteria, we deemed this sufficient evidence for establishing that the data were normally distributed. The differences in numerical variables among the four groups were assessed via one-way ANOVA combined with Tukey post-hoc testing, or via the Kruskal–Wallis and Mann–Whitney U test. For testing the difference in frequencies between study groups, Chi-squared or Fisher's exact test was performed. In order to identify factors independently associated

**Table 1 Demographic characteristics and systemic diseases of subjects with XFG, POAG, XFS and NC.**

| Characteristic | XFG ($n^* = 47$) | POAG ($n^* = 43$) | $p^b$ | NC ($n^* = 53$) | $p^c$ | XFS ($n^* = 22$) | $p^d$ | $p^a$ |
|---|---|---|---|---|---|---|---|---|
| Age, mean ± SD (y) | 73.28 ± 9.00 | 66.33 ± 7.77 | 0.001* | 73.45 ± 9.49 | 1.000 | 75.45 ± 7.03 | 0.746 | <0.001* |
| Male gender, $n$ (%) | 30 (63.8) | 23 (53.5) | 0.319 | 33 (62.3) | 0.871 | 10 (45.5) | 0.150 | 0.418 |
| DM, $n$ (%) | 12 (25.5) | 15 (34.9) | 0.334 | 15 (28.3) | 0.755 | 6 (27.3) | 0.878 | 0.792 |
| HbA1c (%) | 5.72 ± 0.50 | 6.02 ± 0.89 | 0.424 | 5.91 ± 0.79 | 0.656 | 5.83 ± 0.83 | 0.908 | 0.439 |
| Blood glucose | 5.69 ± 1.67 | 6.28 ± 2.82 | 0.617 | 6.25 ± 2.29 | 0.504 | 5.61 ± 1.38 | 0.997 | 0.251 |
| SH, $n$ (%) | 36 (76.6) | 27 (62.8) | 0.153 | 38 (71.7) | 0.577 | 16 (72.7) | 0.473 | 0.539 |
| MS, $n$ (%) | 4 (8.5) | 1 (2.3) | 0.201 | 2 (3.8) | 0.319 | 1 (4.5) | 0.554 | 0.553 |
| Arrhythmia, $n$ (%) | 12 (25.5) | 5 (11.6) | 0.092 | 12 (22.6) | 0.736 | 4 (18.2) | 0.500 | 0.383 |
| CAB or VS, $n$ (%) | 10 (21.3) | 4 (9.3) | 0.117 | 4 (7.5) | 0.048* | 3 (13.6) | 0.449 | 0.184 |
| AAA, $n$ (%) | 3 (6.4) | 2 (4.7) | 0.720 | 0 (0) | 0.062 | 0 (0) | 0.546 | 0.226 |
| ACD, $n$ (%) | 3 (6.4) | 2 (4.7) | 0.720 | 3 (5.7) | 0.879 | 1 (4.5) | 0.761 | 0.982 |

Notes:
  XFG, exfoliative glaucoma; POAG, primary open-angle glaucoma; NC, normal controls; XFS, controls with exfoliation syndrome; n*, number of patients; y, years; DM, diabetes mellitus; SH, systemic hypertension; MS, myocardial stroke; CAB, coronary artery bypass; VS, vascular surgery; AAA, abdominal aortic aneurysm; ACD, acute cerebrovascular disease.
  * Statistically significant $p$ values.
  [a] Between all groups.
  [b] XFG vs. POAG.
  [c] XFG vs. NC.
  [d] XFG vs. XFS.

with XFG, multivariate logistic regression was performed. Models were constructed for the following inter-group comparisons: XFG vs. POAG, XFG vs. NC and XFG vs. XFS. Model variables were selected via the "Enter" method, and VIF collinearity was examined (all variables with VIF > 5 were eliminated from the primary model). Correlation between ophthalmic clinical parameters in XFG, POAG and XFS eyes at presentation and HS and CS concentrations in serum, as well as correlation between HbA1c and blood glucose and HS and CS concentrations was tested by examining the Spearman's correlation coefficient. Diagnosis screening tests (sensitivity, specificity) and ROC curve analysis have been applied in the determination of cut-off values for analyzed parameters.

All statistical methods were significant at $p \leq 0.05$. Statistical analysis was performed in IBM SPSS ver. 21.0.

## RESULTS

### Demographic characteristics

The study sample comprised of 165 subjects, in 47 and 43 of which XFG and POAG were newly diagnosed, respectively, while 22 patients had XFS and 53 individuals served as NC. The patients' demographic characteristics and the prevalence of systemic diseases in the sample are shown in Table 1. The mean age in the XFG, POAG, XFS and NC group was 73.28 ± 9.00, 66.33 ± 7.77, 75.45 ± 7.03 and 73.45 ± 9.49 years, respectively, whereby the age difference was statistically significant for XFG vs. POAG ($p = 0.001$) only. The age of subjects with XFM (XFG and XFS groups) was 73.97 ± 8.43 and was statistically significantly ($p < 0.001$) higher than that (70.26 ± 9.41 years) of non-XFM subjects (POAG and NC groups combined). Participants assigned to the XFG, POAG and NC groups were

**Table 2 Clinical features and indices of glaucoma severity in XFG and POAG eyes at presentation.**

| Characteristic | XFG ($n^*$ = 76) | POAG ($n^*$ = 86) | $p$ |
|---|---|---|---|
| BCVA | | | <0.001* |
| Med (min–max) | 0.70 (0.008–1.0) | 1.0 (0.03–1.0) | |
| LP, $n$ (%) | 2 (1.8) | 1 (0.7) | |
| NLP, $n$ (%) | 3 (2.6) | 0 (0) | |
| IOP (mmHg) | | | <0.001* |
| mean ± SD | 32.4 ± 10.1 | 28.1 ± 4.7 | |
| Hodapp, $n$ (%) | | | 0.046* |
| Early | 31 (40.8) | 53 (61.7) | |
| Moderate | 16 (21.1) | 15 (17.4) | |
| Advanced | 19 (25.0) | 13 (15.1) | |
| Without visual field | 10 (13.1) | 5 (5.8) | |
| MD, med (min–max) | −5.26 (−1.58 to −28.27) | −4.13 (−1.39 to −31.02) | 0.024* |
| PSD, med (min–max) | 4.05 (1.55–12.43) | 3.02 (1.46–14.9) | 0.128 |
| vC/D, med (min–max) | 0.6 (0.45–1.0) | 0.5 (0.45–1.0) | 0.042* |

**Notes:**
BCVA, best-corrected visual acuity; LP, light perception; NLP, no light perception; IOP, intraocular pressure; MD, mean deviation; PSD, pattern standard deviation; vC/D, vertical cup to disc ratio; $n^*$, number of eyes.
* Statistically significant $p$ values.

predominantly male (at 63.8%, 53.5% and 62.3%, respectively) while XFS subjects were predominantly female (54.5%, $p = 0.150$).

## Systemic diseases

With the exception of history of coronary artery bypass or vascular surgery, prevalence of systemic diseases in the studied groups was not statistically significantly different. As shown in Table 1, while a higher percentage of patients in the XFG group reported a history of coronary artery bypass or vascular surgery compared with other groups, this difference was statistically significant for XFG vs. NC only (21.3% vs. 7.5%, $p = 0.048$).

## Ophthalmic characteristics

The examination was performed on 330 eyes, whereby 76 (23.0%) eyes pertained to newly diagnosed XFG, 86 (26.1%) to newly diagnosed POAG and XFS was noted in 44 (13.3%) eyes, while in the remaining 124 (37.6%) eyes, XFS or any type of glaucoma was absent, which thus served as normal control group.

The IOP in the eyes affected by XFG was higher than the IOP in the POAG eyes, whereby the difference in the mean IOP was statistically significant (32.4 ± 10.1 mm Hg vs. 28.1 ± 4.7 mm Hg, $p < 0.001$). The IOP in the XFS group (16.3 ± 2.6 mm Hg) was higher than in the NC group (15.7 ± 2.1 mm Hg) but the difference was not statistically significant ($p = 0.093$).

Glaucoma severity indices pertaining to the newly diagnosed XFG and POAG eyes are shown in Table 2. A greater visual field loss expressed through Hodapp classification ($p = 0.046$) and MD ($p = 0.024$) was noted in eyes affected by XFG relative to the POAG eyes. A statistically significantly smaller vC/D was recorded in the POAG group ($p = 0.042$).

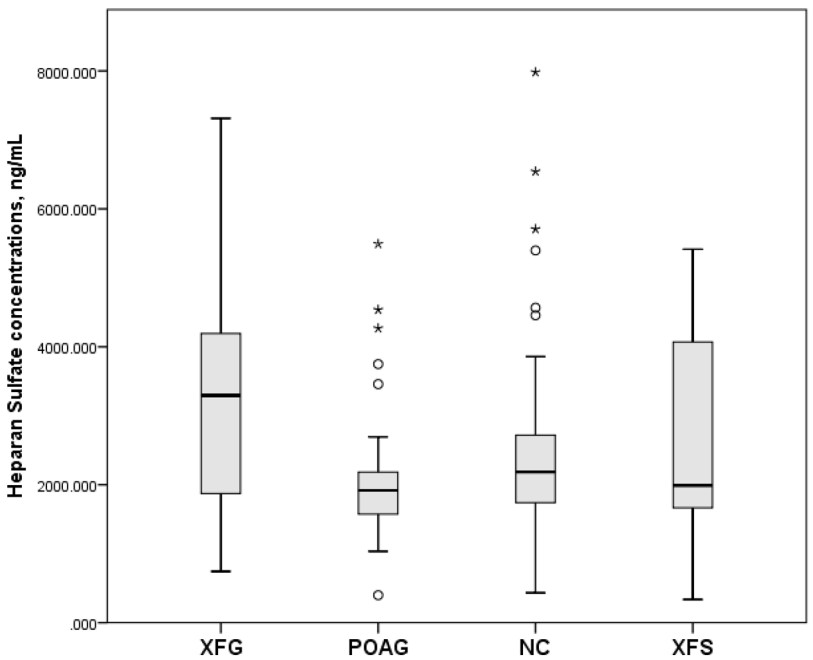

**Figure 1 Serum Heparan Sulfate concentrations, ng/mL in subjects with XFG, POAG, NC, XFS.** Box plots present median, bars are minimum and maximum values, circles are outliers and stars are extreme values. Subjects with XFG, POAG, NC, XFS.

## HEPARAN SULFATE AND CHONDROITIN SULFATE CONCENTRATIONS

Serum human HS concentrations (mean and standard deviation) in the XFG, POAG, NC and XFS groups were 3,189.0 ± 1,473.8 ng/mL, 2,091.5 ± 940.9 ng/mL, 2,543.1 ± 1,397.3 ng/mL and 2,658.2 ± 1,426.8 ng/mL, respectively, whereby only the differences between XFG vs. POAG ($p = 0.001$) and XFG vs. NC ($p = 0.032$) were statistically significant, while that between XFG and XFS groups was not ($p = 0.244$), as shown in Fig. 1.

Serum CS concentrations (mean and standard deviation) in the XFG, POAG, NC and XFS group were 43.9 ± 20.7 ng/mL, 38.5 ± 22.0 ng/mL, 35.8 ± 16.4 ng/mL and 43.3 ± 21.8 ng/mL respectively. However, as shown in Fig. 2, only the difference between XFG and NC was statistically significant ($p = 0.041$), while the difference between XFG and POAG ($p = 0.099$) as well as the XFG and the XFS group was not statistically significant ($p = 0.857$).

When the participants in the XFM (XFG and XFS) group were compared to subjects without XFM (those assigned to the POAG and NC groups), their serum HS concentrations were 3,019.8 ± 1,469.8 ng/mL vs. 2,338.7 ± 1,227.2 ng/mL ($p = 0.006$) (Fig. 3) and CS concentrations were 43.7 ± 20.9 ng/mL vs. 37.1 ± 19.0 ng/mL ($p = 0.026$) (Fig. 4).

In Table 3 we report HS and CS concentrations segregated by gender for the XFG, POAG, XFS and NC groups.

Factors associated with XFG that, according to multivariate logistic regression analysis, distinguish it from POAG, NC and XFS are shown in Table 4.

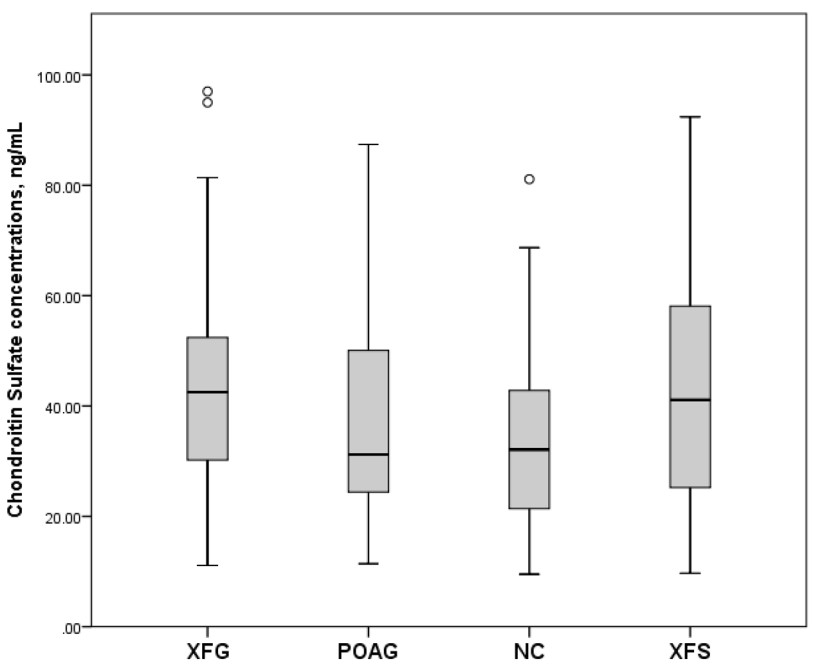

**Figure 2 Serum Chondroitin Sulfate concentrations, ng/mL in subjects with XFG, POAG, NC, XFS.**
Box plots present median, bars are minimum and maximum values, circles are outliers and stars are
extreme values.

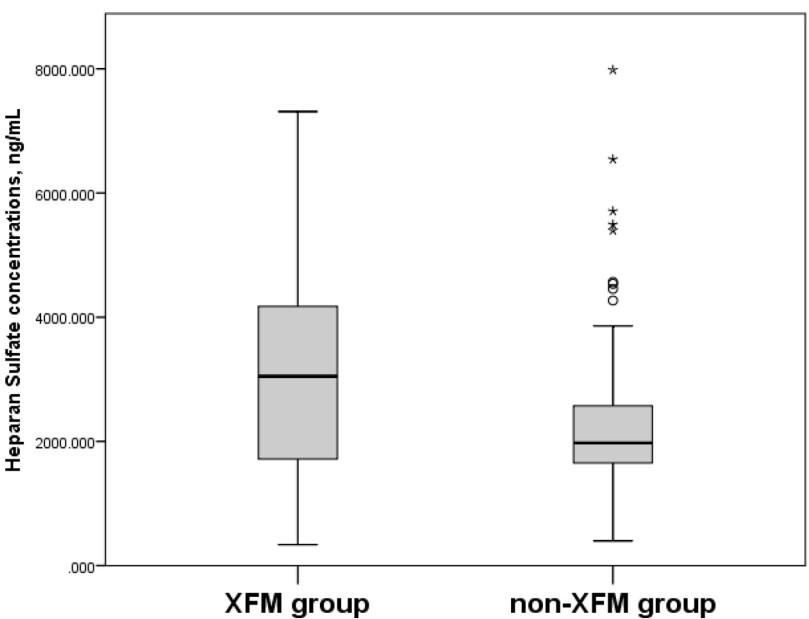

**Figure 3 Serum Heparan Sulfate concentrations, ng/mL in XFM and non-XFM groups.** Box plots
present median, bars are minimum and maximum values, circles are outliers and stars are extreme values. XFM
group: XFG and XFS; Non-XFM group: POAG and NC.

Moreover, as shown in Table 5, the Spearman's correlation coefficient values indicated
absence of correlation between either ophthalmic clinical parameters or indices of glaucoma
severity in XFG, POAG and XFS eyes at presentation and serum HS and CS concentrations.

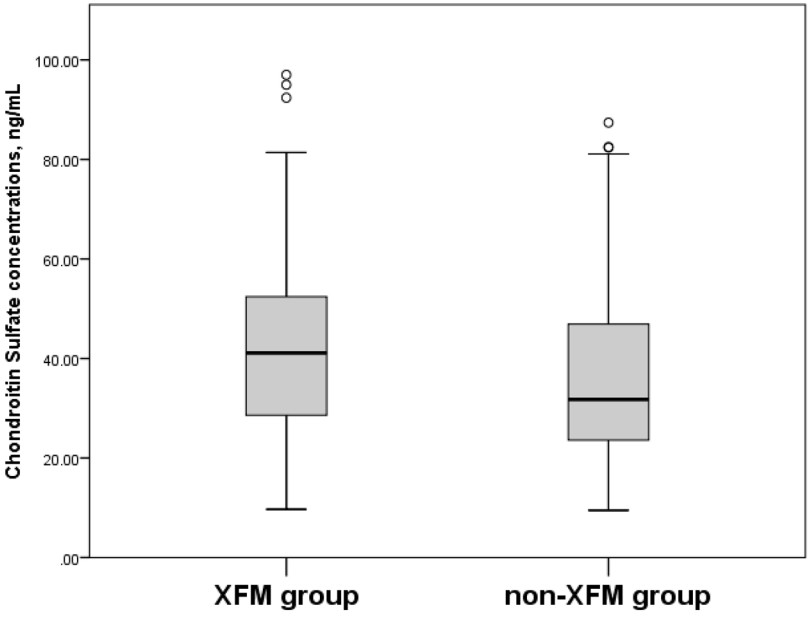

**Figure 4 Serum Chondroitin Sulfate concentrations, ng/mL in XFM and non-XFM groups.** Box plots present median, bars are minimum and maximum values, circles are outliers and stars are extreme values. XFM group: XFG and XFS; Non-XFM group: POAG and NC.

**Table 3 HS and CS concentrations segregated by gender for the XFG, POAG, XFS and NC subjects.**

|   |   | XFG | p | POAG | p | NC | p | XFS | p* |
|---|---|---|---|---|---|---|---|---|---|
| CS | M | 41.98 ± 19.97 | 0.413 | 29.36 ± 14.37 | 0.008* | 34.75 ± 15.68 | 0.660 | 43.30 ± 24.43 | 1.000 |
|   | F | 47.23 ± 22.13 |  | 49.09 ± 24.76 |  | 37.67 ± 17.72 |  | 43.21 ± 21.46 |  |
| HS | M | 3196.7 ± 1562.0 | 0.859 | 1800.4 ± 488.7 | 0.242 | 2579.4 ± 1317.3 | 0.419 | 3172.9 ± 1396.1 | 0.140 |
|   | F | 3175.5 ± 1349.9 |  | 2425.9 ± 1208.6 |  | 2477.7 ± 1520.4 |  | 2229.3 ± 1360.2 |  |

Notes:
HS, heparan sulfate; CS, chondroitin sulfate; M, male gender; F, female gender.
* Statistically significant p values.

Correlation between HbA1c and serum glucose and HS and CS concentrations, as measured by the Spearman's correlation coefficient, is reported in Table 6. In the present study, we evaluated the clinical diagnostic value of HS and CS, in terms of their sensitivity and specificity, in patients with XFG compared to POAG patients, NC and subjects with XFS. When the XFG group was compared to NC, HS had a sensitivity of 60% and specificity of 73% at a cut-off value of 2,688.3 ng/mL. Area under the ROC curve (AUC) was 0.62 (95% CI [0.510–0.741], $p = 0.032$) (Fig. 5). In the comparison between XFG and POAG, HS had a sensitivity of 60% and specificity of 86% at a cut-off value of 2,642.9 ng/mL, and the AUC was 0.70 (95% CI [0.590–0.814], $p = 0.001$). In the comparison of XFG to all remaining groups, HS had a sensitivity of 64% and specificity of 71% at a cut-off value of 2,495.4 ng/mL, with the AUC of 0.65 ($p = 0.003$), and if the group with XFM was compared to the one without XFM, HS had a sensitivity of 59% and specificity of 74% at a cut-off value of 2,495.4 ng/mL, with the AUC of 0.63 (95% CI [0.538–0.721], $p = 0.006$).

**Table 4 Factors associated with XFG vs. POAG, NC, XFS according to multivariate logistic regression analysis.**

| XFG vs. POAG | Multivariate logistic regression analysis | | | |
|---|---|---|---|---|
| | OR, 95% CI OR | $p$ | $p^a$ | $p^b$ |
| Age | 0.989, 0.91–1.07 | 0.787 | / | / |
| BCVA | 307.52, 5.47–172850 | 0.005* | 0.002* | 0.002* |
| IOP | 1.979, 0.85–4.60 | 0.113 | 0.142 | 0.133 |
| Hodapp | 0.431, 0.21–0.87 | 0.018* | 0.018* | 0.019* |
| MD | 0.846, 0.75–0.96 | 0.009* | 0.002* | 0.003* |
| vC/D | 0.953, 0.87–1.05 | 0.320 | 0.164 | 0.198 |
| HS | 0.999, 0.99–1.00 | 0.047* | 0.022* | 0.028* |
| XFG vs. NC | | | | |
| CAB or VS | 6.457, 0.92–45.45 | 0.061 | 0.046* | 0.060 |
| MD | 0.858, 0.31–2.39 | 0.770 | 0.996 | 0.831 |
| vC/D | 0.257, 0.03–2.09 | 0.204 | 0.274 | 0.248 |
| HS | 1.000, 0.99–1.00 | 0.081 | 0.045* | 0.046* |
| CS | 0.855, 0.59–1.24 | 0.095 | 0.091 | 0.065 |
| XFG vs. XFS | | | | |
| vC/D | 1.016, 1.00–1.03 | 0.123 | 0.118 | 0.129 |
| MD | 1.042, 0.28-3.94 | 0.951 | 0.579 | 0.963 |

Notes:
BCVA, best-corrected visual acuity; IOP, intraocular pressure; MD, mean deviation; vC/D, vertical cup to disc ratio; HS, heparan sulfate; CS, chondroitin sulfate; CAB, coronary artery bypass; VS, vascular surgery.
* Statistically significant $p$ values.
$p^a$: adjusted for age and gender.
$p^b$: adjusted for age and gender and presence of diabetes mellitus.

**Table 5 Correlation coefficients between clinical parameters and indices of glaucoma severity at presentation and HS and CS concentrations.**

| Parameters | XFG | | POAG | | XFS | |
|---|---|---|---|---|---|---|
| | HS | CS | HS | CS | HS | CS |
| BCVA | −0.09 | −0.14 | −0.20 | −0.02 | −0.15 | −0.20 |
| IOP | 0.03 | 0.22 | −0.10 | −0.19 | −0.06 | −0.19 |
| Hodapp | −0.10 | 0.11 | −0.13 | −0.12 | / | / |
| MD | −0.17 | −0.30 | 0.13 | 0.30 | 0.24 | 0.42 |
| vC/D | 0.08 | 0.05 | 0.09 | −0.17 | 0.24 | −0.33 |

Note:
No correlation in all examined parameters, Spearman's correlation coefficient BCVA, best-corrected visual acuity; IOP, intraocular pressure; MD, mean deviation; vC/D, vertical cup to disc ratio; HS, heparan sulfate; CS, chondroitin sulfate.

With respect to the diagnostic value of CS, if XFG was compared to NC, its sensitivity and specificity was 60% and 62%, respectively, at a cut-off value of 3.92 ng/mL, while the AUC was 0.62 (95% CI [0.508–0.730], $p = 0.041$) (Fig. 6). When the group with XFM was compared to the one without XFM, CS had a sensitivity of 64% and specificity of 61% at a cut-off value of 3.62 ng/mL, with the AUC of 0.60 (95% CI [0.513–0.690], $p = 0.026$).

If serum HS and CS concentrations were measured simultaneously, the sensitivity and specificity values were 84% and 45%, respectively, when the XFG group was compared to

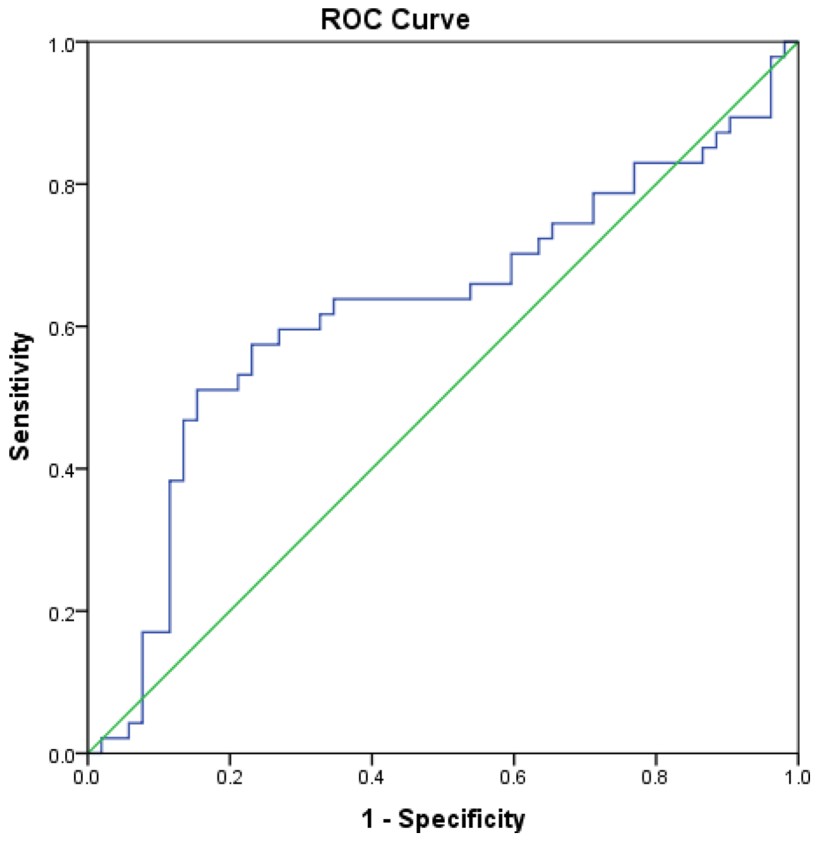

**Figure 5  Area under the ROC curve of HS in patients with XFG compared to NC.** ROC curve drawn for the comparison of serum heparan sulfate concentration between patients with exfoliative glaucoma and normal controls and area under the ROC curve was 0.62 (95% CI [0.510– 0.741]), $p = 0.032$.

**Table 6  Correlation coefficients between HbA1c and blood glucose and HS and CS concentrations.**

| Group | Variable coefficient value ($p$) | |
|---|---|---|
| | Blood glucose | HbA1c |
| XFG | | |
| CS | −0.05 (0.720) | −0.24 (0.105) |
| HS | −0.01 (0.933) | 0.00 (0.998) |
| POAG | | |
| CS | −0.17 (0.273) | −0.28 (0.063) |
| HS | −0.21 (0.168) | −0.07 (0.680) |
| NC | | |
| CS | 0.02 (0.900) | 0.03 (0.819) |
| HS | 0.36 (0.008)* | 0.25 (0.073) |
| XFS | | |
| CS | −0.18 (0.426) | −0.48 (0.024)* |
| HS | −0.18 (0.422) | −0.31 (0.157) |

Notes:
Spearman rank correlation coefficient HS, heparan sulfate; CS, chondroitin sulfate.
* Statistically significant $p$ values.

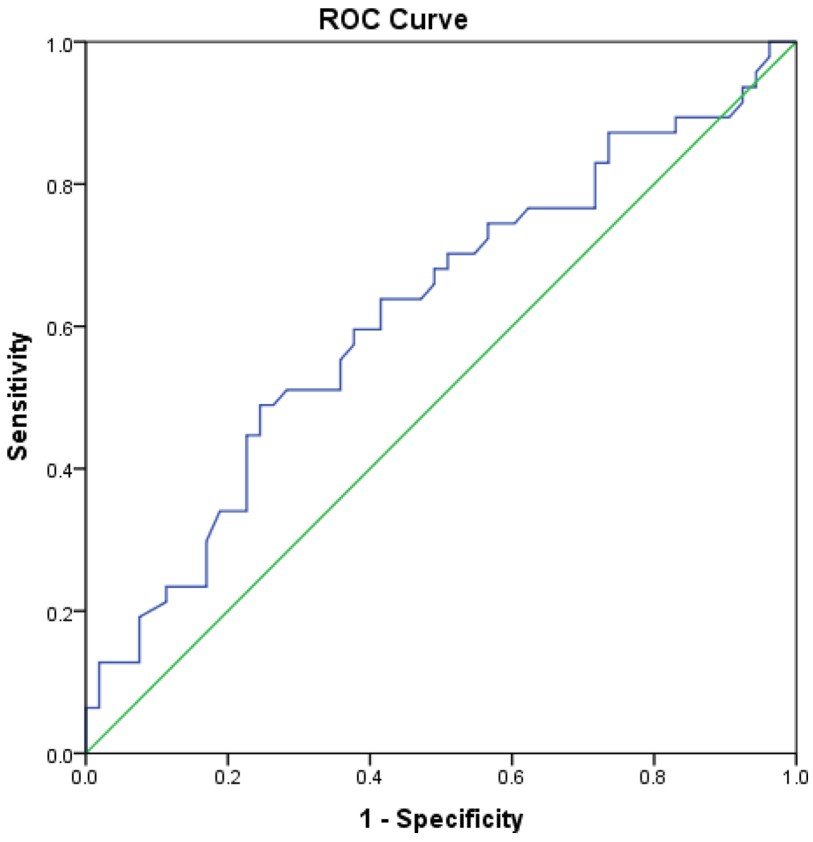

**Figure 6 Area under the ROC curve of CS in patients with XFG compared to NC.** ROC curve drawn for the comparison of serum chondroitin sulfate concentration between patients with exfoliative glaucoma and normal controls and area under the ROC curve was 0.62 (95% CI [0.508–0.730]), $p = 0.041$.

NC, whereas if the group with XFM was compared to that without XFM, a sensitivity of 85% and a specificity of 44% were obtained.

## DISCUSSION

Exfoliation syndrome is an age-related disorder characterized by the pathological accumulation deposition of fibrillar material in multiple tissues, frequently associated with severe secondary open-angle glaucoma (*Puska, 2015*; *Ritch & Schlötzer-Schrehardt, 2001*).

Exfoliative glaucoma typically develops after 60 years of age and, in most cases, significant optic nerve head and visual field damage in at least one eye is already present at the time of diagnosis (*Ritch & Schlötzer-Schrehardt, 2001*). XFG clinical features are distinct from those characterizing POAG (*Gonzalez-Iglesias et al., 2014*).

In the present study, XFG patients were statistically significantly older than those diagnosed with POAG, as their mean age was 7 years greater, but they were of similar age as the XFS participants.

The VA findings yielded by our study indicate a more severe visual loss in newly diagnosed XFG eyes relative to that noted in the POAG group. Compared with the POAG group, higher mean IOP was measured in the XFG eyes that took part in the present study.

Moreover, a significantly greater severity of optic nerve damage was noted in newly diagnosed XFG compared to newly diagnosed POAG, as indicated by vC/D. In addition, the newly diagnosed XFG patients also had significantly more advanced visual field changes than those newly diagnosed with POAG.

As early diagnosis is key to preventing visual impairment in glaucoma, especially in XFG which has aggressive course, extensive research efforts have recently been dedicated to glaucoma biomarker discovery in bodily fluids that would permit disease identification.

In principle, a biomarker is an indicator of a biochemical feature or facet that can be used to diagnose or monitor the progress of a disease (*Ross et al., 2005*). Thus, an agent of measurable entity with high sensitivity and specificity that accurately predicts the presence, progression or absence of a disease would be an "ideal" biomarker. Hence, in XFS subjects, it is critical that a biomarker can accurately and reliably predict progression from syndrome to glaucoma, as well as allow identification of individuals at high risk of progressive damage. Given that XFS is a systemic disease, serum or plasma would be ideal bodily fluids for identification of potential biomarkers. Thus far, systemic XFM has never been diagnosed in the absence of intraocular manifestations (*Ritch & Schlötzer-Schrehardt, 2001*).

Human serum or plasma serve as typical clinical specimens, since both are accessible and convenient for use in clinical trials. It has been demonstrated that the low molecular weight fraction of human serum or plasma provides a rich source of potential biomarkers pertaining to diseases generated through enzymatic cleavage (*Anderson et al., 2004*; *Adkins et al., 2002*). The aim of the present study was to investigate with relatively simple method, patients' serum samples, in order to examine potential biomarkers (GAG concentrations, CS and HS) in newly diagnosed untreated XFG and compare their levels with those pertaining to newly diagnosed untreated POAG, as well as NC and subjects with XFS. To the best of our knowledge, no prior studies on this topic have been conducted to date.

Chondroitin sulfate and HS are two distinct classes of GAGs. Both are distributed on the surface of virtually all cells and throughout most extracellular matrices. Consequently, CS and HS are the major GAGs in the blood, whereby other serum GAGs include keratin sulfate and hyaluronan (*Lu et al., 2010*). GAGs such as CS and HS in human serum or plasma carry important biological information and their variations in human serum or plasma are vital for investigating and monitoring certain disease conditions, such as cancer, rheumatoid arthritis and DM (*Li et al., 2017*; *Pothacharoen et al., 2006*; *Lamari et al., 2006*; *Komosińska-Vassev et al., 2005*). Therefore, evaluating GAG presence and quantity variations has a great potential for disease diagnosis and prognosis (*Wei et al., 2011*). Nevertheless, no standardized methods exist for serum/plasma GAG isolation and quantification (*Lu et al., 2010*). The approach based on a sandwich ELISA adopted in the present study was easy to perform, and it yielded reliable and reproducible quantitative results.

In the current study, HS serum concentration in newly diagnosed XFG patients was significantly higher relative to that in newly diagnosed POAG patients and NC. While elevated concentrations were also noted for the XFG group relative to the subjects

with XFS, the difference was not statistically significant. Similarly, serum CS concentrations in newly diagnosed XFG patients were elevated compared to the NC, as well as POAG and XFS subjects; however, only the XFG vs. NC difference was statistically significant.

As already mentioned, XFS is a generalized disorder of the ECM characterized by production and progressive accumulation of XFM deposits in tissues of the anterior segment, as well a systemic disorder. These findings suggest that ocular XFS is part of a general disorder of the ECM and that patients with XFM may suffer from increased comorbidity. As XFS/XFG results in excessive pathological ECM deposition, this increases GAG, as well as heparan and CS levels.

When the CS values were examined with respect to gender, they were higher in women relative to men in the XFG, POAG and NC groups, but the difference was statistically significant in the POAG group only. Exfoliation syndrome was the only group in which male subjects had negligibly higher CS values. On the other hand, HC concentration values were higher among men in XFG, NC and XFS groups, but the difference relative to women was not statistically significant. Slightly higher HC values were obtained for women in the POAG group.

The multivariate analysis findings revealed that, GAG concentrations (especially HS) were associated with XFG. In addition, statistically significant differences in the serum concentrations of GAGs, HS and CS were noted between XFM patients (XFG and XFS group) and subjects without XFM (those assigned to the POAG and NC groups).

The total amount of GAGs in serum was previously found to be closely related to diabetic pathology, advanced diabetic complications in particular (*Komosińska-Vassev et al., 2005*). However, in our study, presence of DM did not influence the results, and the differences among groups were not statistically significant. Moreover, in line with the results reported by *Shingleton, Heltzer & O'Donoghue (2003)*, DM was slightly less frequent in the exfoliation group relative to the non-exfoliation group. Similarly, *Musch et al. (2012)* found diabetes significantly more frequent among patients with POAG (18.2%) vs. patients with XFG (3.5%). In an earlier study, *Konstas et al. (1998)* noted a lower prevalence of diabetes in patients with XFG requiring surgery than in those with POAG. Thus, in multivariate logistic regression analyses, we adjusted for presence of DM. On the other hand, blood glucose and HbA1c values were not statistically significantly different across groups. No link between blood glucose and HbA1c and HS and CS values was noted in the XFG group, while in the NC group HS and blood glucose were in the moderately positive statistically significant correlation, whereas in the XFS group HbA1c was in a negative moderate correlation with CS.

Identification of potential XFG biomarkers from bodily fluids has been the subject of extensive research. For example, *McNally & O'Brien (2014)* highlighted some important findings yielded by employing metabolomics and proteomics strategies in the search for possible biomarkers of XFG from anterior lens capsule, blood or aqueous humor in exfoliation glaucoma patients. Similarly, *Kamel, Bourke & O'Brien (2018)* conducted a literature review in order to present an up-to-date list of clinical and laboratory-based biomarkers relating to XFS and XFG.

In prior studies in this field, analyses were time consuming and complex in most cases, involving analysis possible biomarkers from the anterior lens capsules, TM and the aqueous humor. *Ghanem, Arafa & El-Baz (2011)* analyzed the aqueous humor composition in XFS/XFG patients because all ocular tissues involved are bathed by the aqueous humor and should therefore be influenced by the factors contained therein. However, the practical utility of these findings is limited, as aqueous humor cannot be easily obtained in routine clinical practice.

*Gonzalez-Iglesias et al. (2014)* conducted a comparative proteomic study involving serum of patients with POAG and XFG to identify a candidate panel of glaucoma biomarkers for the clinical prediction, prognosis, diagnosis and monitoring of POAG and XFG cases.

The novel contribution of the present study stems from attempting to establish a link between HS and CS serum concentrations with ophthalmic clinical parameters in XFG, XFS and POAG eyes, which was not supported by the research findings. Moreover, no correlation was noted between HS and CS serum concentrations and glaucomatous damage in newly diagnosed XFG and POAG cases, expressed via glaucoma severity indices, such as vC/D, Hodapp Classification and MD.

Our aim was to identify a biomarker that would be specific to patients with XFG. Our results as well as findings reported by other authors indicate that, in XFG, significant optic nerve head and visual field damage is already present at the time of diagnosis (*Ritch & Schlötzer-Schrehardt, 2001*; *Konstas et al., 2006*). Consequently, in the involved eye(s), optic nerve head damage and visual filed deterioration are frequently more severe in XFG compared to POAG patients (*Ritch, 2001*). Owing to these facts, we attempted to discover serum biomarkers that would prompt a patient to seek ophthalmologist consultation, if indicated by the blood analysis results, in order to obtain a timely XFG diagnosis and start appropriate therapy. This is highly relevant, given that older individuals tend to visit general practitioners more frequently than ophthalmologists. As a part of the present investigation, we evaluated the clinical diagnostic value of HS and CS and, according to our findings, we established that HS is an adequate diagnostic test for comparing XFG to NC and all other control groups, whereas CS is an adequate diagnostic test for comparing XFG with NC. Moreover, these tests, when performed separately, can adequately distinguish subjects with XFM from those without XFM. Clearly, performing both tests would be advantageous, as this would increase sensitivity. It should be noted that the results reported here were affected by a small sample size, as noted in study limitations.

When interpreting the results yielded by our investigation, it is important to note some study limitations. Specifically, as the study aim was to determine the time of the XFG and POAG occurrence, we did not select POAG patients that were age- and sex-matched to the XFG cohort. The existence of systemic diseases was established through interviews, as well as via a detailed review of medical documentation. Thus, it needs to be emphasized that systemic diseases were not clinically determined in our study sample. It is also noteworthy that we might have failed to detect some subclinical cases of XFS, which would be diagnosable by histological methods only. Moreover, DM patients were not excluded from the sample, as already explained, and we adjusted for presence of DM in

multivariate logistic regression analyses. Our findings confirmed absence of statistically significant differences between groups, indicating that inclusion of DM patients should not have influenced our results. A further limitation of the study is a small sample size.

## CONCLUSIONS

In conclusion, our findings revealed greater HS and CS concentrations in newly diagnosed XFG patients and XFS subjects compared to those without XFM. Heparan sulfate is an adequate diagnostic test for comparing XFG to NC and all other control groups, whereas CS is an adequate diagnostic test for comparing XFG with NC.

As this work, to the best of our knowledge, marks the first attempt to evaluate HS and CS serum concentrations in both XFG and XFS subjects, this data and ongoing research in this field will continue to improve our understanding of XFS/XFG as a systemic disorder. Implications of HS and CS in the pathophysiology of XFS and glaucoma should be studied further. Serum or plasma is easily accessible and should thus be explored as rich sources of potential disease biomarkers. Further research should aim to identify XFG biomarkers that could be utilized in routine blood analysis tests, aiding in timely disease diagnosis.

### Funding
This investigation was supported by the Ministry of Education, Science and Technological Development of the Republic of Serbia (Grant Nos. 175042 and 175087). The funders had no role in study design, data collection and analysis, decision to publish, or preparation of the manuscript.

### Grant Disclosures
The following grant information was disclosed by the authors:
This investigation was supported by the Ministry of Education, Science and Technological Development of the Republic of Serbia: 175042 and 175087.

### Competing Interests
The authors declare that they have no competing interests.

### Author Contributions
- Vesna D. Maric conceived and designed the experiments, performed the experiments, analyzed the data, contributed reagents/materials/analysis tools, prepared figures and/or tables, authored or reviewed drafts of the paper.
- Marija M. Bozic authored or reviewed drafts of the paper, approved the final draft.
- Andja M. Cirkovic analyzed the data, prepared figures and/or tables.
- Sanja Dj Stankovic performed the experiments, contributed reagents/materials/ analysis tools.
- Ivan S. Marjanovic prepared figures and/or tables.
- Anita D. Grgurevic authored or reviewed drafts of the paper, approved the final draft.
## Human Ethics

The following information was supplied relating to ethical approvals (i.e., approving body and any reference numbers):

This human study is a part of the doctoral dissertation and this dissertation was approved by the Ethics Committee of the Faculty of Medicine, University of Belgrade, record number 29/III-3, on March 13th 2017.

Thus, as a part of the documentation submission to the Ethics Committee, all patient examinations were described in detail, along with all assessments, analyses and diagnostic procedures that were included and described in this study. All study methods and procedures were approved by the Ethics Committee of the Faculty of Medicine, University of Belgrade. This work does not contain additional experiments that are not presented in the dissertation.

## Data Availability

The raw data is available in the Supplementary File.

## Supplemental Information

Supplemental information for this article can be found online at http://dx.doi.org/10.7717/peerj.6920#supplemental-information.

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
