# Peer review of "Serum heparan sulfate and chondroitin sulfate concentrations in patients with newly diagnosed exfoliative glaucoma"

_PeerJ, doi:10.7717/peerj.6920_

## Round 0.1 · original submission · Major Revisions

Thank you for submitting your manuscript to PeerJ. Based on the comments of three reviewers and my own reading of your manuscript I invite you to resubmit after making major revisions. All three reviewers were positive about your paper and have made a number of suggestions that I invite you to address in a revised submission and response letter. Reviewer 3 recommends that you shorten your introduction. I would recommend organizing the introduction into multiple paragraphs and consider if any content should be reduced. Reviewer 3 also recommends focusing your discussion more on your findings. Please consider the organization of your discussion and how it addresses both your findings and their relationship to the literature. This would help you to address and respond to reviewer 3’s suggestion that the paragraph stretching from lines 337-364 is not necessary.

I look forward to receiving your revised manuscript. Please be sure to respond to all reviewer comments in a response letter with your resubmission.

Reviewer 1 ·

Basic reporting

line 25: typo: exraocular - should be extraocular
line 158: typo: filed - should be visual field

no further comment

Experimental design

Yes: original primary research within Aims and Scope of the journal
Yes: Research question well defined, relevant & meaningful; fills a knowledge gap
Yes: Methods described with sufficient detail & information to replicate pending answers to below:

Suggestion for improvement: Overall great study
line 106: For the authors - please clarify: was POAG defined present with the typical glaucomatous disc WITH OR WITHOUT glaucoma visual field changes (or is it and/or glaucoma visual field changes, as is written. Same question for PXG, line 108-109.

The reason to ask for this important clarification is "The vast majority of of patients with POAG have disc changes or disc and visual field changes, but there may be rare cases where there may be early visual field changes before there are detectable changes to the optic nerve." Preferred Practice Pattern: Primary Open Angle Glaucoma; American Academy of Ophthalmology 2016.

Please clarify: were patients with diabetic retinopathy excluded?

Validity of the findings

Results:
line 234: what was the p value for IOP in the XFS group vs NC group?

line 255: while that between XFG and POAG was NOT significant (rather than borderline significant (p = 0.099)

limitations: consider including small sample size

consider further commentary on how Diabetes could have affected the results and average HgA1c if available

Additional comments

Overall this is a good study and idea, that could generate future studies and investigation into therapeutic measures and early disease detection. Please see the comments above

·

Basic reporting

In my opinion the overall strength of manuscript can be improved by explaining the differences between the XFG, XFS and POAG in the introduction for common audience.
How many subjects did authors excluded based on exclusion criteria?
Most of the documents in supplemental folder are written in Serbian language. The documents must be translated and verified.
Measurement range and detection limit for Human chondroitin sulfate should be written in methods section.
In line 195 change “Tuckey” to “Tukey”

Experimental design

Even though XFS is a systemic disease, XFG and POAG are ocular diseases so shouldn’t tears be more ideal biological fluid to look for biomarkers?
Did the authors observed any gender differences in the HS and CS expression levels?
Did the authors agree that ocular characteristics like IOP, BCVA seems to be better biomarkers than CS and HS? If yes, then what is the rational behind looking for HS and CS in serum?

Validity of the findings

In the discussion the authors note the importance of finding sensitive and specific biomarkers to accurately predict the presence and progression of disease however in the current study they did not provide the specificity and sensitivity values of HS and CS. Without good sensitivity and specificity, the significant differences observed by the authors cannot be used as a good biomarker.
In Fig’s 1-4 if the bars represent min and max values how can they have outliers and extreme values shouldn’t they be the parts of bars they are min and max values?

·

Basic reporting

No coments

Experimental design

No comment

Validity of the findings

No comment

Additional comments

Dear authors,
Thanks for the opportunity to revise your manuscript. I found it very interesting. As I work and investigate in exfoliation since many years ago, I think we need to find biomarkers to identify the disease and to identify the patients in which the disease can led to blindness.
Before acceptance certain changes must be done.
The introduction is too long, please keep around 40%.
In lines 193-195, about statistics, please explain. It seems like you have used both parametric and no parametric models. Why? In which cases?
The discussion is not so focused in your findings. Lines 334-336, I cannot understand what you mean. The whole paragraph (lines 337-364) has no meaning.
Please try to hypothesize the relationship between your findings and exfoliation. Common chemical pathways? Do you think that heparan and condroitin sulfate come from the eye? If not, from which part of the body do you think they are coming?

---

## Round 0.2 · Minor Revisions

Thank you for your thorough revision and response to all reviewer comments. The reviewers all agree that the paper is greatly improved. I have recommended minor revisions so that you can address the apparent disagreement between reviewer 1 and 2 about the proper biomarkers for XFG. Please either make the change requested by reviewer 1 or explain why this change should not be made.

I would also recommend combining the first two paragraphs of the revised introduction (lines 52-57) and re-wording to reduce repetitive use of the word glaucoma, if possible.

I do not plan to send a revision back out to the reviewers so a decision can be made promptly once these final issues are addressed.

Reviewer 1 ·

Basic reporting

line 72, GAGs, are (not as)

lines 393-412: much improved, recommend tightening up and organizing (ie, no need write, on the other hand, in other words..)

lines 466-468: IOP and BCVA are not better biomarkers for XFG; IOP and BCVA are routine parameters checked in standard eye exams. I recognize that the authors addresses IOP and BCVA due to a reviewers comments; however, IOP and BCVA are nor better biomarkers for XFG; I recommend remove lines 466-468

Experimental design

good

Validity of the findings

can improve clarity/organization of conclusions, tighten up

Additional comments

Great work and much improved manuscript. This is novel work and can stimulate further investigation.

·

Basic reporting

I have no further comments.

Experimental design

I have no further comments.

Validity of the findings

I have no further comments.

Additional comments

The authors answered all questions. So, I recommend the publication of the manuscript.

·

Basic reporting

No comments

Experimental design

No comments

Validity of the findings

No comments

Additional comments

I'm pleased by the changes made in the manuscript.

---

## Round 0.3 · accepted · Accept

Thank you for addressing the remaining reviewer comments and for your revised submission. I am happy to now accept your manuscript for publication in PeerJ.

I would still recommend combining the first two sentences of the introduction (lines 52 to 57) into one paragraph. If you choose to do so you can make that edit in the final production draft.

You will be given the option to make the reviews of your manuscript available to readers. Please consider doing so as this review record can be a great resource for readers of your paper and contributes to more transparent science.

Thank you again for your contribution.